# EFFECTIVE DIMENSION OF MACHINE LEARNING MODELS

## ABSTRACT

Making statements about the performance of trained models on tasks involving new data is one of the primary goals of machine learning, i.e., to understand the generalization power of a model. Various capacity measures try to capture this ability, but usually fall short in explaining important characteristics of models that we observe in practice. In this study, we propose the local effective dimension as a capacity measure which seems to correlate well with generalization error on standard data sets. Importantly, we prove that the local effective dimension bounds the generalization error and discuss the aptness of this capacity measure for machine learning models.

## 1 INTRODUCTION

The essence of successful machine learning lies in the creation of a model that is able to learn from data and apply what it has learned to new, unseen data (Goodfellow et al., 2016). The latter ability is termed the *generalization performance* of a machine learning model and has proven to be notoriously difficult to predict a priori (Zhang et al., 2021). The relevance of generalization is rather straightforward: if one already has insight on the performance capability of a model class, this will allow for more robust models to be selected for training and deployment. But how does one begin to analyze generalization without physically training models and assessing their performance on new data thereafter? This age-old question has a rich history and is largely addressed through the notion of capacity. Loosely speaking, the capacity of a model relates to its ability to express a variety of functions (Vapnik et al., 1994). The higher a model's capacity, the more functions it is able to fit. In the context of generalization, many capacity measures have been shown to mathematically bound the error a model makes when performing a task on new data, i.e. the *generalization error* (Vapnik & Chervonenkis, 1971; Liang et al., 2019; Bartlett et al., 2017). Naturally, finding a capacity measure that provides a tight generalization error bound, and in particular, correlates with generalization error across a wide range of experimental setups, will allow us to better understand the generalization performance of machine learning models.

Interestingly, through time, proposed capacity measures have differed quite substantially, with trade-offs apparent among each of the current proposals (Jiang et al., 2019). The perennial VC dimension has been famously shown to bound the generalization error, but it does not incorporate crucial attributes, such as data potentially coming from a distribution, and ignores the learning algorithm employed which inherently reduces the space of models within a model class that an algorithm has access to (Vapnik et al., 1994). Arguably, one of the most promising contenders for capacity which attempts to incorporate these factors are norm-based capacity measures, which regularize the margin distribution of a model by a particular norm that usually depends on the model's trained parameters (Bartlett et al., 2017; Neyshabur et al., 2017b; 2015). Whilst these measures incorporate the distribution of data, as well as the learning algorithm, the drawback is that most depend on the size of the model, which does not necessarily correlate with the generalization error in certain experimental setups (Zhang et al., 2021).

To this end, we present the *local effective dimension* which attempts to address these issues. By capturing the redundancy of parameters in a model, the local effective dimension is modified from (Berezniuk et al., 2020; Abbas et al., 2021) to incorporate the learning algorithm employed, in addition to being scale invariant and data dependent. The key results from our study can be summarized as follows:

Table 1: Overview of established capacity measures and desirable properties. The first property is whether the measure can be mathematically related to the generalization error via an upper bound. The second states whether this bound is good in practice, i.e., that the measure correlates with the generalization error in various experimental setups, such as (Zhang et al., 2021). Scale invariance corresponds to the measure being insensitive to inconsequential transformations of the model, such as multiplying a neural network's weights by a constant. Data and training dependence refers to a measure accounting for data drawn from a distribution and the learning algorithm employed. Finite data merely implies that the measure can handle finite data. Lastly, efficient evaluation refers to the possibility of estimating the capacity measure in polynomial time (in the number of data).

| | VC-dimension | Rademacher complexity | Margin-based | Norm-based | Sharpness-based | Local ED |
|---|---|---|---|---|---|---|
| 1. Generalization bound | ✓ | ✓ | ✓ | ✓ | ✓ | ✓ |
| 2. Correlation with generalization | ✗ | ✗ | ✗ | ✗ | ✗ | ✓ |
| 3. Scale invariant | ✗ | ✓ | ✗ | ✗ | ✗ | ✓ |
| 4. Data dependent | ✗ | ✓ | ✓ | ✓ | ✓ | ✓ |
| 5. Training dependent | ✗ | ✗ | ✓ | ✓ | ✓ | ✓ |
| 6. Finite data | ✗ | ✓ | ✓ | ✓ | ✓ | ✓ |
| 7. Efficient evaluation | ✗ | ✗ | ✓ | ✓ | ✓ | ✓ |

- The local effective dimension, unlike its predecessors, includes training dependence as well as other desirable properties summarized in Table 1.

- We prove that the local effective dimension bounds the generalization error of a trained model with finite data (see Theorem 4.1).

- The local effective dimension largely depends on the Fisher information, which is often approximated in practice (Kunstner et al., 2019). We rigorously quantify the sensitivity of the local effective dimension when evaluated with an approximated Fisher information (see Proposition 3.2).

- Lastly, we empirically show that the local effective dimension correlates well with generalization error in various experimental setups using standard data sets. The local effective dimension is found to decrease in line with the generalization error as a network increases in size. Similarly, the measure increases in line with the generalization error when models are trained on randomized training labels.

## 2 PRELIMINARIES

In this section, we provide an overview of relevant literature and a concise introduction to generalization error bounds and the Fisher information.

### 2.1 RELATED WORK

We briefly discuss relevant capacity measures proposed in literature, but defer to (Jiang et al., 2019) for a more comprehensive overview. Given a model class, Vapnik *et al.* showed that the VC dimension can provide an upper bound on generalization error (Vapnik et al., 1994). While this was a crucial first step in using capacity to understand generalization, the VC dimension rests on unrealistic assumptions, such as access to infinite data, and ignores things like training dependence and the fact that data, more reasonably, comes from a distribution (Holden & Niranjan, 1995). The closely-related Rademacher complexity relaxes some of the assumptions made on the model class, but still suffers similar issues to the VC dimension (Yin et al., 2019; Wang et al., 2018). Since then, a myriad of capacity measures aiming to circumvent these problems and provide tighter generalization error bounds, have been proposed. Margin-based capacity measures stemmed from the work of Vapnik and Chervonenkis in 1974 who pointed out that generalization error bounds based on the VC dimension may be significantly enhanced in the case of linear classifiers that produce large margins. In (Bartlett et al., 1998), it was shown that the phenomenon where boosting models (no matter how large you make them) do not overfit data, could also be explained by the large margins these boosting models achieved. Since the

grand mystery in modern deep learning can be characterized by the same phenomenon – extremely large overparameterized neural networks that seemingly do not overfit data – it seems natural to try extend the idea of margin bounds to these model families. Moreover, margin-based approaches allow us to leverage the fact that learning algorithms, like gradient decent, produce classifiers with large margins on training data.[1] Unfortunately, looking at margins in isolation does not say much about the performance of deep neural networks on unseen data. There have been recent investigations on how to add a normalization such that margin-based measures become informative. Most of these proposals involve the incorporation of the Lipschitz constant of a network, which is simply the product of the spectral norms of the weight matrices (Bartlett et al., 2017). These normalized margin-based techniques gave rise to norm-based capacity measures which appear promising, however, it is still unclear how to perform this normalization and often, the normalization depends on some factor that scales with the size of the model, which is undesirable in the case of deep neural networks (Neyshabur et al., 2015).

Another interesting proposal for measuring capacity came about by trying to characterize the local minima achieved by deep networks after training (Keskar et al., 2016; Hochreiter & Schmidhuber, 1997). These so-called sharpness-based measures often depend on the Hessian, which incorporates a notion of curvature at a particular point in the loss landscape. It was believed that sharper minima led to better generalization properties, although this was later shown to be incorrect as sharpness measures were usually not scale invariant and thus, did not correlate well with generalization error in various scenarios (Dinh et al., 2017).

This leads us to the purpose of this study where we introduce and motivate the local effective dimension as a capacity measure. The effective dimension arises from the principle of minimum description length and thus, tries to capture existing redundancy in a statistical model (Berezniuk et al., 2020; Cover & Thomas, 2006; Rissanen, 1996). Redundancy has been widely studied in deep learning through techniques like pruning and model compression (Yeom et al., 2021; Molchanov et al., 2019; Wiedemann et al., 2020; Tung & Mori, 2020; Cheng et al., 2018; 2017; Tishby & Zaslavsky, 2015). Interestingly, attempts to connect redundancy/minimum description to generalization performance have also been studied in (Hinton & van Camp, 1993; Achille & Soatto, 2018; MacKay, 1992), and the idea was used to compare the capacity of quantum and classical machine learning models in (Abbas et al., 2021). We refine the existing definitions of the effective dimension, which in turn leads us to the creation of a local version that conveniently meets the criteria presented in Table 1.

## 2.2 GENERALIZATION ERROR

A typical first step in motivating use for a capacity measure is to prove that it bounds the generalization error. Informally, all generalization error bounds have the same structure

$$\text{generalization error} \leq \text{estimate of error} + \text{complexity penalty},$$

which attempts to relate generalization error to an empirical estimate of the error, plus a complexity penalty captured by the proposed capacity measure. Since empirical estimates correspond to the training error on available data, which can mostly be trained to zero with very deep networks in practice, the capacity term is usually of most relevance. More formally, suppose we are given a hypothesis class, $\mathcal{H}$, of functions mapping from $\mathcal{X}$ to $\mathcal{Y}$, a training set $\mathcal{S}_n = \{(x_1, y_1), \ldots, (x_n, y_n)\} \in (\mathcal{X} \times \mathcal{Y})^n$ where the data pairs $(x_i, y_i)$ are drawn i.i.d. from some unknown joint distribution $p$, and let $\ell : \mathcal{Y} \times \mathcal{Y} \to \mathbb{R}$ be a loss function. The machine learning task is to find a particular hypothesis $h \in \mathcal{H}$ that minimizes the *expected risk*, defined as $R(h) := \mathbb{E}_{(x,y)\sim p}[\ell(h(x), y)]$. Since we only have access to a training set $\mathcal{S}_n$, a good strategy to find the best hypothesis $h \in \mathcal{H}$ is to minimize the so called *empirical risk*, defined as $R_n(h) := \frac{1}{n} \sum_{i=1}^n \ell(h(x_i), y_i)$. The difference between the expected and the empirical risk is known as the *generalization error gap*. This gap gives us an indication as to whether a hypothesis $h \in \mathcal{H}$ will perform well on unseen data, drawn from the unknown joint distribution $p$ (Neyshabur et al., 2017a). Therefore, an upper bound on the quantity

$$\sup_{h \in \mathcal{H}} |R(h) - R_n(h)|, \tag{1}$$

---

[1]A beautiful review of margin bounds is encapsulated in (Anthony & Bartlett, 2009) where lower bounds are proved for certain function classes.

which vanishes as $n$ grows large, is of considerable interest.[2] Capacity measures help quantify the expressiveness and power of $\mathcal{H}$. Thus, the quantity in equation 1 is typically bounded by an expression that depends on some notion of capacity(Vapnik et al., 1994).

## 2.3 FISHER INFORMATION

The Fisher information has many interdisciplinary interpretations (Frieden, 2004). In machine learning, several capacity measures incorporate the Fisher information in different ways (Liang et al., 2019; Tsuda et al., 2004). It is also a crucial quantity in the effective dimension and is thus, briefly introduced here.

Consider a parameterized statistical model $p(x, y; \theta) = p(y|x; \theta)p(x)$ which describes the joint relationship between data pairs $(x, y)$ for all $x \in \mathcal{X}$, $y \in \mathcal{Y}$ and $\theta \in \Theta \subseteq \mathbb{R}^d$. The input distribution, $p(x)$, is a prior distribution over the data and the conditional distribution, $p(y|x; \theta)$ describes the input-output relation generated by the model for a fixed $\theta \in \Theta$. The full parameter space $\Theta$ forms a Riemannian space which gives rise to a Riemannian metric, namely, the Fisher information which we can represent in matrix form

$$F(\theta) = \mathbb{E}_{(x,y) \sim p} \left[ \frac{\partial}{\partial \theta} \log p(x, y; \theta) \frac{\partial}{\partial \theta} \log p(x, y; \theta)^{\mathsf{T}} \right].$$

By definition, the Fisher information matrix is positive semidefinite and hence, its eigenvalues are non-negative. In practical applications where $d$ is typically large, there exists sophisticated techniques to efficiently approximate the Fisher information matrix. This is discussed in Appendix E.3.

## 3 EFFECTIVE DIMENSION

The origin of the effective dimension arose from a simple operational question: Is it possible to quantify the number of parameters that are truly active in a statistical model?[3] In the case of deep neural networks, it has already been shown that many parameters are inactive, inspiring better design techniques (Han et al., 2015). Measuring parameter activeness can be made mathematically precise with tools from statistics and information theory. In particular, the effective dimension unites the principle of minimum description length with the Kolmogorov complexity of a model (Rissanen, 1996; Cover & Thomas, 2006). We introduce the global effective dimension here and refer the interested reader to (Berezniuk et al., 2020; Abbas et al., 2021) for more details.

### 3.1 GLOBAL EFFECTIVE DIMENSION

To shorten notation we write

$$\kappa_{n,\gamma} = \frac{\gamma n}{2\pi \log n}, \tag{2}$$

for $n \in \mathbb{N}$, which represents the number of data samples available, and a constant $\gamma \in (\frac{2\pi \log n}{n}, 1]$.

**Definition 3.1.** *The* global effective dimension *of a statistical model* $\mathcal{M}_\Theta := \{p(\cdot, \cdot; \theta) : \theta \in \Theta \subset \mathbb{R}^d\}$ *with respect to* $n \in \mathbb{N}$ *and* $\gamma \in (\frac{2\pi \log n}{n}, 1]$*, is defined as*

$$d_{n,\gamma}(\mathcal{M}_\Theta) := \frac{2 \log \left( \frac{1}{V_\Theta} \int_\Theta \sqrt{\det \left( \mathrm{id}_d + \kappa_{n,\gamma} \bar{F}(\theta) \right)} \, d\theta \right)}{\log \kappa_{n,\gamma}}, \tag{3}$$

*where* $V_\Theta := \int_\Theta d\theta \in \mathbb{R}_+$ *is the volume of the parameter space and* $\kappa_{n,\gamma}$ *is defined in equation 2. The matrix* $\bar{F}(\theta) \in \mathbb{R}^{d \times d}$ *is the normalized Fisher information matrix defined as*

$$\bar{F}_{ij}(\theta) := d \frac{V_\Theta}{\int_\Theta \mathrm{tr}(F(\theta)) d\theta} F_{ij}(\theta),$$

*where* $F(\theta) \in \mathbb{R}^{d \times d}$ *denotes the Fisher information matrix of* $p(\cdot, \cdot; \theta)$.

---

[2]We assume that the loss function is Lipschitz continuous which implies that equation 1 vanishes as $n \to \infty$.

[3]A parameter is considered active if it has a sufficiently large influence on the outcome of its statistical model, i.e. varying the parameter changes the model.

For conciseness, we simply denote the global effective dimension as $d_{n,\gamma}$. The global effective dimension converges to the maximal rank of the Fisher information matrix $\bar{r} := \max_{\theta \in \Theta} r_\theta \in \{1, 2, \ldots, d\}$ in the limit of $n \to \infty$, where $r_\theta$ denotes the rank of $F(\theta)$. Thus, it often makes sense to standardize the measure by looking at the *normalized effective dimension*, denoted by $\bar{d}_{n,\gamma} = d_{n,\gamma}/d$, which gives us a proportion of active parameters relative to the total number of parameters in the model.

We prove that the global effective dimension is continuous as a function of the Fisher information matrix. Since the Fisher information is typically approximated in practice, such a statement is relevant to ensure small deviations in the Fisher information do not exacerbate possible deviations in the global effective dimension (see Section 5 for more details).[4]

**Proposition 3.2** (Continuity of the effective dimension). *Let $n \in \mathbb{N}$, $\gamma \in (\frac{2\pi \log n}{n}, 1]$, and consider two statistical models $\mathcal{M}_\Theta$ and $\mathcal{M}'_\Theta$ with $\Theta \subset \mathbb{R}^d$ and corresponding Fisher information matrices $F$ and $F'$, respectively. Then,*

$$|d_{n,\gamma}(\mathcal{M}_\Theta) - d_{n,\gamma}(\mathcal{M}'_\Theta)| \leq C_d \left( \frac{1}{\phi(F)} + \frac{1}{\phi(F')} \right) \max_{\theta \in \Theta} \left\| \sqrt{\bar{F}(\theta)} - \sqrt{\bar{F}'(\theta)} \right\|$$
$$+ \frac{2\psi(F) + 2\psi(F')}{\log \kappa_{n,\gamma}},$$

*where $C_d$ is a dimensional constant, $\kappa_{n,\gamma}$ is defined in equation 2, $\phi(F) := \frac{1}{V_\Theta} \int_\Theta \sqrt{\det(\bar{F}(\theta))} d\theta$, and*

$$\psi(F) = \max \left\{ \log \left( \frac{1}{V_\Theta} \int_\Theta \sqrt{\det(\mathrm{id}_d + \bar{F}(\theta))} d\theta \right), -\log \left( \frac{1}{V_\Theta} \int_\Theta \sqrt{\det(\bar{F}(\theta))} d\theta \right) \right\}.$$

The proof is given in Appendix B. Proposition 3.2 is informative for statistical models $\mathcal{M}_\Theta$ and $\mathcal{M}'_\Theta$ with corresponding Fisher information matrices $F$ and $F'$ such that $\phi(F) > 0$ and $\phi(F') > 0$, respectively. This unavoidable consequence is due to $\lim_{n \to \infty} d_{n,\gamma}(\mathcal{M}_\Theta) = \bar{r} = \max_{\theta \in \Theta} \mathrm{rank}(F(\theta))$ and $\lim_{n \to \infty} d_{n,\gamma}(\mathcal{M}_\Theta) = \bar{r}' = \max_{\theta \in \Theta} \mathrm{rank}(F'(\theta))$. Hence, we see that when $\bar{r} \neq \bar{r}'$, the effective dimension is not continuous as $n \to \infty$. This is consistent with Proposition 3.2 where in the case of $\bar{r} < d$ or $\bar{r}' < d$, we have $\phi(F) = 0$ or $\phi(F') = 0$, respectively.

**Remark 3.3** (Stabilized computation of the effective dimension). *For large $d$, sufficiently large $n$ and models with a full rank Fisher information matrix, the effective dimension is of order $d$. This implies that $\det(\mathrm{id}_d + \kappa_{n,\gamma}\bar{F}(\theta))$ is exponentially large in $d$, which makes direct calculation of the effective dimension via equation 3 numerically challenging when large models are considered. This can be circumvented by rewriting the effective dimension as*

$$d_{n,\gamma}(\mathcal{M}_\Theta) = \frac{2}{\log \kappa_{n,\gamma}} \log \left( \frac{1}{V_\Theta} \int_\Theta \exp \left( \frac{1}{2} \log \det \left( \mathrm{id}_d + \kappa_{n,\gamma}\bar{F}(\theta) \right) \right) d\theta \right).$$

*Noting that*

$$\frac{1}{2} \log \det \left( \mathrm{id}_d + \kappa_{n,\gamma}\bar{F}(\theta) \right) = \frac{1}{2} \mathrm{tr} \log \left( \mathrm{id}_d + \kappa_{n,\gamma}\bar{F}(\theta) \right)$$
$$= \frac{1}{2} \sum_{i=1}^d \log \left( 1 + \kappa_{n,\gamma}\lambda_i(\bar{F}(\theta)) \right) =: z(\theta), \quad (4)$$

*where $\lambda_i(\bar{F}(\theta))$ denotes the $i$-th eigenvalue of $\bar{F}(\theta)$. The quantity $z(\theta)$ can be computed without any under- or overflow problems for large $n$ and $d$. Choosing $\zeta = \max_{\theta \in \Theta} z(\theta)$ then gives*

$$d_{n,\gamma}(\mathcal{M}_\Theta) = \frac{2\zeta}{\log \kappa_{n,\gamma}} + \frac{2}{\log \kappa_{n,\gamma}} \log \left( \frac{1}{V_\Theta} \int_\Theta \exp \left( z(\theta) - \zeta \right) d\theta \right),$$

*which is a numerically stable expression for the effective dimension.*

---

[4]This proposition also holds for the local effective dimension introduced in Definition 3.4.

## 3.2 LOCAL EFFECTIVE DIMENSION

While the global effective dimension has nice properties, an important aspect to note is that it incorporates the full parameter space $\Theta$. In practice, however, training models with a learning algorithm inherently restricts the space of parameters that a model truly has access to. Once a model is trained, only a fixed parameter set $\theta^\star \in \Theta$ is considered, which is chosen to minimize a certain loss function. This leads us to the introduction of the local effective dimension which accounts for dependence on the training algorithm. To achieve this, we define an $\epsilon$-ball around a fixed parameter set $\theta^\star \in \Theta \subset \mathbb{R}^d$ for $\epsilon > 0$ as

$$\mathcal{B}_\epsilon(\theta^\star) := \left\{ \theta \in \Theta : \|\theta - \theta^\star\| \leq \epsilon \right\},$$

with a volume $V_\epsilon := \int_{\mathcal{B}_\epsilon(\theta^\star)} \mathrm{d}\theta$.

**Definition 3.4.** *The* local effective dimension *of a statistical model* $\mathcal{M}_\Theta := \{ p(\cdot, \cdot; \theta) : \theta \in \Theta \}$ *around* $\theta^\star \in \Theta$ *with respect to* $n \in \mathbb{N}$, $\gamma \in (\frac{2\pi \log n}{n}, 1]$, *and* $\epsilon > 1/\sqrt{n}$ *is defined as*

$$d_{n,\gamma}(\mathcal{M}_{\mathcal{B}_\epsilon(\theta^\star)}) = \frac{2 \log \left( \frac{1}{V_\epsilon} \int_{\mathcal{B}_\epsilon(\theta^\star)} \sqrt{\det \left( \mathrm{id}_d + \kappa_{n,\gamma} \bar{F}(\theta) \right)} \, \mathrm{d}\theta \right)}{\log \kappa_{n,\gamma}},$$

*for* $\kappa_{n,\gamma}$ *given by equation 2. The matrix* $\bar{F}(\theta) \in \mathbb{R}^{d \times d}$ *is the normalized Fisher information matrix defined as*

$$\bar{F}_{ij}(\theta) := d \frac{V_\epsilon}{\int_{\mathcal{B}_\epsilon(\theta^\star)} \mathrm{tr}(F(\theta)) \mathrm{d}\theta} F_{ij}(\theta),$$

*where* $F(\theta) \in \mathbb{R}^{d \times d}$ *denotes the Fisher information matrix of* $p(\cdot, \cdot; \theta)$.

For ease of notation, we denote the local effective dimension as $d_{n,\gamma,\epsilon}$. From Definition 3.4, we immediately see that the local effective dimension is scale invariant as it depends on the normalized Fisher information matrix, as well as training dependent, since the training determines $\theta^\star$. Via its dependence on the Fisher information, the local effective dimension also incorporates an assumed distribution for the data and is built for finite data, as summarized in Table 1. Proposition 3.2 further proves that the local effective dimension is continuous in the Fisher information matrix.

The computationally dominant part in evaluating the local effective dimension is the calculation of the Fisher information matrix. Luckily, this is a well-studied problem with existing proposals for efficient evaluation (Kunstner et al., 2019; Martens & Grosse, 2015). Since we only require the eigenvalues of the Fisher matrix for the local effective dimension, we can further exploit these Fisher approximations and do not need to store a $d \times d$ matrix (see Section E.3 for more details). Additionally, the integral over the $\epsilon$-ball can be evaluated efficiently with Monte-Carlo type methods.

To complete the criteria from Table 1, it remains to show that the local effective dimension bounds and correlates with the generalization error, which is illustrated next.

## 4 GENERALIZATION AND THE LOCAL EFFECTIVE DIMENSION

Understanding the role of the local effective dimension in the context of generalization requires a rigorous relationship to be defined. We demonstrate this relationship through the generalization error, bounded by the local effective dimension.

### 4.1 GENERALIZATION ERROR BOUND

Consider machine learning models described by stochastic maps, parameterized by some $\theta \in \Theta$ and a loss function as a mapping $\ell : \mathrm{P}(\mathcal{Y}) \times \mathrm{P}(\mathcal{Y}) \to \mathbb{R}$ where $\mathrm{P}(\mathcal{Y})$ denotes the set of distributions on $\mathcal{Y}$. The following regularity assumption on the model $\mathcal{M}_\Theta := \{ p(\cdot, \cdot; \theta) : \theta \in \Theta \}$ is assumed:

$$\Theta \ni \theta \mapsto p(\cdot, \cdot; \theta). \tag{5}$$

is $M_1$-Lipschitz continuous w.r.t. the supremum norm.

**Theorem 4.1.** *Let $\Theta = [-1, 1]^d$ and consider a statistical model $\mathcal{M}_\Theta := \{p(\cdot, \cdot; \theta) : \theta \in \Theta\}$ satisfying equation 5 such that $\bar{F}(\theta)$ has full rank for all $\theta \in \Theta$, and $\|\nabla_\theta \log \bar{F}(\theta)\| \leq \Lambda$ for some $\Lambda \geq 0$ and all $\theta \in \Theta$. Furthermore, let $\ell : \mathrm{P}(\mathcal{Y}) \times \mathrm{P}(\mathcal{Y}) \to [-B/2, B/2]$ for $B > 0$ be a loss function that is Lipschitz continuous with constant $M_2$ in the first argument with respect to the total variation distance. Then, there exists a dimensional constant $c_d$ such that for $\theta^\star \in \Theta$, $n \in \mathbb{N}$, $\gamma \in (\frac{2\pi \log n}{n}, 1]$, and $\epsilon > 1/\sqrt{n}$ we have*

$$\mathbb{P}\left(\sup_{\theta \in \mathcal{B}_\epsilon(\theta^\star)} |R(\theta) - R_n(\theta)| \geq \frac{4M\epsilon}{\sqrt{\kappa_{n,\gamma}}}\right) \leq c_d(1 + \epsilon\Lambda)^d \cdot \kappa_{n,\gamma}^{\frac{d_{n,\gamma,\epsilon}}{2}} \exp\left(-\frac{16\pi M^2 \epsilon^2 \log n}{B^2 \gamma}\right),$$

*where $M = M_1 M_2$, $\kappa_{n,\gamma}$ is defined in equation 2, and $d_{n,\gamma,\epsilon}$ is the local effective dimension $d_{n,\gamma}(\mathcal{M}_{\mathcal{B}_\epsilon(\theta^\star)})$.*

Theorem 4.1 assumes that the loss function is Lipschitz continuous. This excludes some popular loss functions such as the relative entropy. Hence, in Appendix C, we extend Theorem 4.1 to include loss functions that are log-Lipschitz.[5]

## 4.2 PROOF OF THEOREM 4.1

Let $\mathcal{N}^{\mathcal{B}_\epsilon(\theta^\star)}(r)$ denote the number of boxes of side length $r$ required to cover the set $\mathcal{B}_\epsilon(\theta^\star)$ – the length being measured with respect to the metric $\bar{F}_{ij}(\theta^\star)$.

**Lemma 4.2.** *Under the assumption of Theorem 4.1, we have for any $\xi \in (0, 1)$*

$$\mathbb{P}\left(\sup_{\theta \in \mathcal{B}_\epsilon(\theta^\star)} |R(\theta) - R_n(\theta)| \geq \xi\right) \leq 2\mathcal{N}^{\mathcal{B}_\epsilon(\theta^\star)}\left(\frac{\xi}{4M}\right) \exp\left(-\frac{n\xi^2}{2B^2}\right).$$

*Proof.* If we replace the full parameter space, $\Theta$, by the relevant reduced space, $\mathcal{B}_\epsilon(\theta^\star)$, the proof of this lemma follows directly from (Abbas et al., 2021, Lemma 2 in the Supplementary Information) if we set $\alpha = 1$. $\qquad\square$

**Lemma 4.3.** *Under the assumption of Theorem 4.1, there exists a dimensional constant $c_d$ such that*

$$\mathcal{N}^{\mathcal{B}_\epsilon(\theta^\star)}\left(\frac{\epsilon}{\sqrt{\kappa_{n,\gamma}}}\right) \leq c_d(1 + \epsilon\Lambda)^d \cdot \kappa_{n,\gamma}^{\frac{d_{n,\gamma,\epsilon}}{2}}.$$

*Proof.* If we choose $\Theta = \mathcal{B}_1(\theta^\star)$ instead of $[-1, 1]^d$, we can then rescale $\mathcal{B}_\epsilon(\theta^\star) \to \mathcal{B}_1(\theta^\star)$, $\bar{F}(\theta) \to \bar{F}(\epsilon\theta)$, $1/\sqrt{n} \to 1/(\epsilon\sqrt{n})$, and $r \to r/\epsilon$.[6] In other words, the number of balls of radius $r$ needed to cover $\mathcal{B}_\epsilon(\theta^\star)$ is equal to the number of balls of radius $r/\epsilon$ needed to cover $\mathcal{B}_1(\theta^\star)$. Then, constants $c_d$ and $\hat{c}_d$ exist such that

$$\mathcal{N}^{\mathcal{B}_\epsilon(\theta^\star)}(r) = \mathcal{N}^{\mathcal{B}_1(\theta^\star)}(r/\epsilon)$$

$$\leq \hat{c}_d(1 + c_d\epsilon\Lambda)^d \frac{1}{V_1} \int_{\mathcal{B}_1(\theta^\star)} \sqrt{\det\left(\mathrm{id}_d + \frac{\epsilon^2}{r^2}\bar{F}(\epsilon\theta)\right)} \mathrm{d}\theta$$

$$= \hat{c}_d(1 + c_d\epsilon\Lambda)^d \frac{1}{V_\epsilon} \int_{\mathcal{B}_\epsilon(\theta^\star)} \sqrt{\det\left(\mathrm{id}_d + \frac{\epsilon^2}{r^2}\bar{F}(\theta)\right)} \mathrm{d}\theta.$$

Hence, choosing $(\epsilon/r)^2 = \kappa_{n,\gamma}$ gives

$$\mathcal{N}^{\mathcal{B}_\epsilon(\theta^\star)}\left(\frac{\epsilon}{\sqrt{\kappa_{n,\gamma}}}\right) \leq c_d(1 + c_d\epsilon\Lambda)^d \cdot \kappa_{n,\gamma}^{\frac{d_{n,\gamma,\epsilon}}{2}},$$

which proves the assertion of the lemma. $\qquad\square$

---

[5]We say that a function $f$ is log-Lipschitz with constant $L$ if $|f(x) - f(y)| \leq L|x - y| \log(\mathrm{e} + 1/|x - y|)$.
[6]Recall that by assumption of Theorem 4.1, we have $\epsilon > 1/\sqrt{n}$.

Thanks to Lemmas 4.2 and 4.3 we can deduce Theorem 4.1. For $\xi = 4M\epsilon/\sqrt{\kappa_{n,\gamma}}$ we find

$$\mathbb{P}\left(\sup_{\theta \in \mathcal{B}_\epsilon(\theta^\star)} |R(\theta) - R_n(\theta)| \geq 4M\epsilon/\sqrt{\kappa_{n,\gamma}}\right) \leq 2\mathcal{N}^{\mathcal{B}_\epsilon(\theta^\star)}\left(\epsilon/\sqrt{\kappa_{n,\gamma}}\right) \exp\left(-\frac{16\pi M^2 \epsilon^2 \log n}{B^2 \gamma}\right)$$

$$\leq 2c_d(1 + \epsilon\Lambda)^d \cdot \kappa_{n,\gamma}^{\frac{d_{n,\gamma,\epsilon}}{2}} \exp\left(-\frac{16\pi M^2 \epsilon^2 \log n}{B^2 \gamma}\right),$$

which completes the proof.[7] □

### 4.3 REMARKS ON THE GENERALIZATION ERROR BOUND

Ideally, the generalization bound from equation 4.1 should be non-vacuous. This occurs if the right-hand side is smaller than one, or equivalently, when the logarithm of the right-hand side is negative. Table 2 demonstrates that a choice for $\gamma \in (\frac{2\pi \log n}{n}, 1]$ such that the bound remains non-vacuous, is reasonable in practical settings where we set $\gamma = 0.003$, but could become vacuous in deeper regimes. For more details, see Appendix D.

## 5 EMPIRICAL RESULTS

In this section, we perform experiments to verify whether the local effective dimension captures the true behaviour of generalization error in various regimes. We use standard fully-connected feedforward neural networks with two hidden layers and vary the model size by altering the number of neurons in the hidden layers. All training was conducted with batched stochastic gradient descent, with experimental setups identical to those of (Liang et al., 2019).[8] The details can be found in Appendix E.

We consider both shallow and deep regimes by training models on MNIST and CIFAR10 data sets, with the latter requiring far more parameters for the training to converge to zero error. Within these regimes, we conduct two experiments respectively: first, we incrementally increase the model size, train to zero error and calculate the local effective dimension, along with the generalization error; second, we replicate the experiment from (Zhang et al., 2021) by fixing the model size and randomizing the training labels by an increasing proportion, training to zero error and calculating the local effective dimension and generalization error. In all calculations, we perform simulations using the K-FAC approximation of the Fisher information from (Martens & Grosse, 2015). K-FAC crucially allows computation of the local effective dimension in very large parameter spaces and we further exploit the block structure of this approximation for computation of the eigenvalues of the Fisher information matrix (George, 2021).

Regardless of the regime and particular experiment conducted, the local effective dimension seems to move in line with the generalization error. In Figures 1(a) and 2(a), we see this for an increasing model size shown on the horizontal axes (with notably much larger models used to learn the CIFAR10 data set). As the models get larger, they are able to perform better on the learning task at hand and their generalization error declines accordingly, as does the (normalized) local effective dimension. The error bars represent the standard deviation around the mean of 10 independent training runs. A lower normalized local effective dimension as the model size increases, intuitively implies increasing redundancy, as also suggested in (Frankle & Carbin, 2018), and motivates pruning techniques (Karnin, 1990).

In Figures 1(b) and 2(b), we fix the model size to $d \approx 10^5$ and $d \approx 10^7$ respectively. Here, the horizontal axis marks the level at which the labels of the training data have been randomized. We begin at 20% randomization to 100% in increments of 20%. At all points, we train to zero training loss or terminate at 600 epochs and plot the resulting normalized local effective dimension and generalization error. Naturally, the generalization performance worsens as we randomize more labels since the network is fitting more and more noise that has been artificially introduced. Interestingly, the local effective dimension captures this behaviour too - increasing with the generalization error -

---

[7]The full rank assumption of the Fisher information matrix in Theorem 4.1 can be relaxed following the ideas from (Abbas et al., 2021, Remark 2).

[8]We include results with the ADAM optimiser in the appendix.

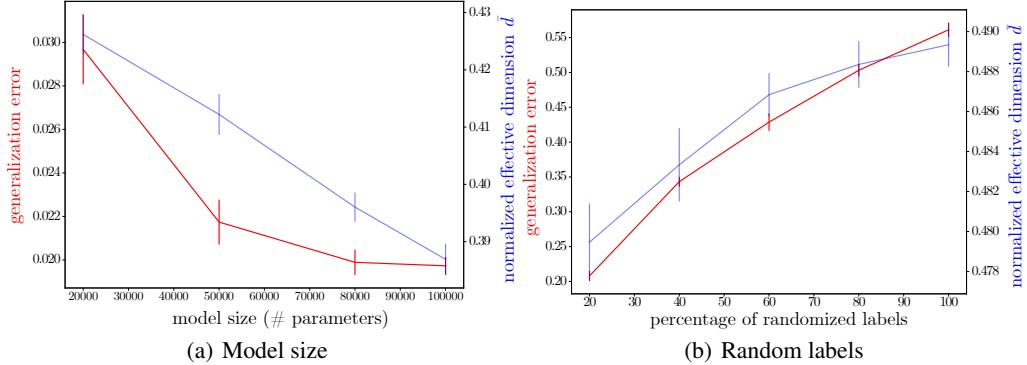

(a) Model size          (b) Random labels

Figure 1: **MNIST** (a) Normalized local effective dimension and generalization error plotted over different model sizes (standard feedforward networks with two hidden layers and varying number of neurons). The parameter $n$ is fixed to equal the size of the training set, i.e. $n = 60000$. (b) Normalized local effective dimension and generalization error over different percentages of randomized labels on the training data. Here, the model size is fixed to $d \approx 10^5$.

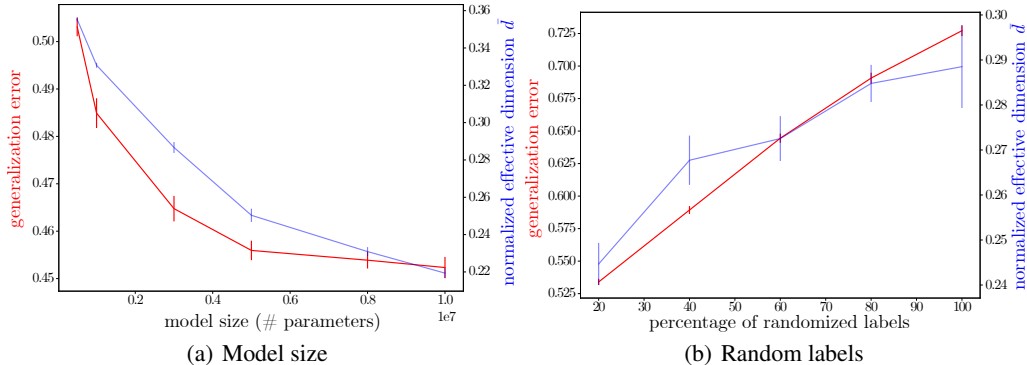

(a) Model size          (b) Random labels

Figure 2: **CIFAR10** (a) Normalized local effective dimension and generalization error plotted over different model sizes. The number of parameters required to train CIFAR10 is far greater than the number of training samples ($n = 50000$). We observe that the local effective dimension moves in line with the declining generalization error as the model is made larger. (b) Normalized local effective dimension and generalization error over different percentages of randomized labels on the training data. Here, the model size is fixed to $d \approx 10^7$.

indicating that more and more parameters need to become "active" to fit this noise. This result is independent of the regime, deep or shallow.

## 6 DISCUSSION

Whilst the search for a good capacity measure continues, we believe that the local effective dimension serves as a promising candidate. Besides being able to correlate with the generalization error in different experiments, the local effective dimension incorporates data, training and does not rest on unrealistic assumptions. It's intuitive interpretation as a measure of redundancy in a model, along with proof of a generalization error bound, suggests that the local effective dimension can explain the performance of machine learning models in various regimes. Investigation into the tightness of the generalization bound, in particular for specific model architectures and in the deep learning regime (where bounds are typically vacuous), would be beneficial in further understanding the local effective dimension's connection to generalization. Additionally, empirical analyses involving bigger models, different data sets and other training techniques/optimizers could shed more light on the practical usefulness of this promising capacity measure.

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

## A  BENEFITS OF THE LOCAL EFFECTIVE DIMENSION

When deciding what is a good measure of capacity for a model, in particular for deep neural networks which are notoriously difficult to understand in a generalization context, it is helpful to check whether the capacity measure satisfies certain criteria which we highlight in Table 1. The first criterion is whether the measure can be mathematically related to the generalization error via an upper bound. This is the main contribution of our work, where we essentially show that the local effective dimension can indeed bound the generalization error. The question of whether one

can obtain tighter generalization bounds using the effective dimension, for specific models, is left for future research. However, there are several interesting pieces of work that could be relevant for this direction, such as (Pennington & Worah, 2018) who investigate the spectrum of the Fisher information for a single layer neural network with infinite width. Since the effective dimension depends largely on the eigenvalues of the Fisher matrix, this would be a convenient place to start this investigation. Additionally, the work in (Pennington & Worah, 2018) shows that a single linear layer network produces a Fisher information spectrum which converges to a Marchenko-Pastur distribution in the infinite width limit. In this setting, the generalization bound based on the effective dimension reduces to something quite trivial which depends primarily on $n$, since the number of data determines how many eigenvalues are counted in the local effective dimension. We hope that future studies can improve the bound we present in this work, as we do not yet explore any optimality results.

The second criterion for a well-proposed capacity measure tries to address whether the generalization bound using the capacity measure is actually good in practice, i.e., that the measure correlates with the generalization error in various experimental setups, such as (Zhang et al., 2021). Through our numerical experiments in Section 5, we answer this with an affirmative answer.

Another crucial property for capacity is scale invariance, which corresponds to the measure being insensitive to inconsequential transformations of the model, such as multiplying a neural network's weights by a constant. Since the local effective dimension is a function of the Fisher information, which is inherently scale invariant, this requirement is naturally accounted for in the local effective dimension.

A good capacity measure should also account for data and training dependence, i.e. the fact that data is drawn from a distribution and one imposes a learning algorithm. Once again, the Fisher information incorporates the data distribution, and the purpose of the localization of the effective dimension is to account for training dependence.

A capacity measure should also be realistic in the sense that it should allow for finite data, which is always the case in practice. The local effective dimension not only allows for finite data, but is structured for this realistic purpose to include the amount of data available as a resolution parameter. This creates a beautiful operational meaning for the local ED that depends on the amount of data one has in practice.

Lastly, a capacity measure should be computationally efficient to evaluate (in polynomial time in the number of data). Thanks to various approximations of the Fisher information, this too is possible for the local effective dimension and is explained in Appendix E.3.

## B  PROOF OF PROPOSITION 3.2

We denote the maximal rank of $F$ and $F'$ by $\bar{r}$ and $\bar{r}'$, respectively, and define the function

$$f(t) := \frac{1}{V_\Theta} \int_\Theta \det \left( \mathrm{id}_d / \sqrt{\kappa_{n,\gamma}} + \underbrace{t \sqrt{\bar{F}(\theta)} + (1-t)\sqrt{\bar{F}'(\theta)}}_{=:G_t} \right) \mathrm{d}\theta \,. \tag{6}$$

We consider a modified version of the effective dimension, defined as

$$\tilde{d}_{n,\gamma}(F) := \frac{2\log f(1)}{\log \kappa_{n,\gamma}} + \bar{r} \qquad \text{and} \qquad \tilde{d}_{n,\gamma}(F') := \frac{2\log f(0)}{\log \kappa_{n,\gamma}} + \bar{r}' \,.$$

The triangle inequality then gives

$$|d_{n,\gamma}(F) - d_{n,\gamma}(F')| \le |d_{n,\gamma}(F) - \tilde{d}_{n,\gamma}(F)| + |\tilde{d}_{n,\gamma}(F) - \tilde{d}_{n,\gamma}(F')| + |\tilde{d}_{n,\gamma}(F') - d_{n,\gamma}(F')| \,. \tag{7}$$

We next bound all the three terms. For the first and the last one, recall (Abbas et al., 2021, Supplementary Information, Section 2) that

$$d_{n,\gamma}(F) \le \bar{r} + \frac{2}{\log \kappa_{n,\gamma}} \log \left( \frac{1}{V_\Theta} \int_\Theta \sqrt{\det(\mathrm{id}_d + \bar{F}(\theta))} \mathrm{d}\theta \right)$$

and for $\mathcal{A} := \{\theta \in \Theta : r_\theta = \bar{r}\}$

$$d_{n,\gamma}(F) \geq \bar{r} + \frac{2}{\log \kappa_{n,\gamma}} \log \left( \frac{1}{V_\Theta} \int_\Theta \sqrt{\det(\bar{F}(\theta))} d\theta \right).$$

Recalling that

$$\psi(F) = \max \left\{ \log \left( \frac{1}{V_\Theta} \int_\Theta \sqrt{\det(\mathrm{id}_d + \bar{F}(\theta))} d\theta \right), -\log \left( \frac{1}{V_\Theta} \int_\Theta \sqrt{\det(\bar{F}(\theta))} d\theta \right) \right\}$$

gives $|d_{n,\gamma}(F) - \tilde{d}_{n,\gamma}(F)| \leq \frac{2\psi(F)}{\log \kappa_{n,\gamma}}$ and $|d_{n,\gamma}(F') - \tilde{d}_{n,\gamma}(F')| \leq \frac{2\psi(F')}{\log \kappa_{n,\gamma}}$. It thus remains to bound middle term in equation 7.

To do so note that

$$|\log f(1) - \log f(0)| \leq \int_0^1 \frac{|f'(t)|}{f(t)} dt.$$

We can bound the numerator of the integral as

$$
\begin{aligned}
|f'(t)| &\leq \frac{1}{V_\Theta} \int_\Theta \left| \frac{\mathrm{d}}{\mathrm{d}t} \det(\mathrm{id}_d/\sqrt{\kappa_{n,\gamma}} + G_t) \right| d\theta \\
&\leq \frac{1}{V_\Theta} \int_\Theta C_{\theta,d} \left\| \sqrt{\bar{F}(\theta)} - \sqrt{\bar{F}'(\theta)} \right\| d\theta \\
&\leq C_d \max_{\theta \in \Theta} \left\| \sqrt{\bar{F}(\theta)} - \sqrt{\bar{F}'(\theta)} \right\|,
\end{aligned}
\tag{8}
$$

where the constant $C_d$ depends on $d$, $\|\sqrt{\bar{F}}\|^{d-1}$, and $\|\sqrt{\bar{F'}}\|^{d-1}$.

Using the fact that $A \mapsto (\det A)^{1/d}$ is concave on the space of Hermitian positive definite matrices gives

$$
\begin{aligned}
\det \left( \mathrm{id}_d/\sqrt{\kappa_{n,\gamma}} + G_t \right) &\geq \left( t \det \left( \mathrm{id}_d/\sqrt{\kappa_{n,\gamma}} + G_1 \right)^{1/d} + (1-t) \det \left( \mathrm{id}_d/\sqrt{\kappa_{n,\gamma}} + G_0 \right)^{1/d} \right)^d \\
&\geq t^d \det \left( \mathrm{id}_d/\sqrt{\kappa_{n,\gamma}} + G_1 \right) + (1-t)^d \det \left( \mathrm{id}_d/\sqrt{\kappa_{n,\gamma}} + G_0 \right).
\end{aligned}
$$

Hence we have $f(t) \geq t^d f(1) + (1-t)^d f(0)$. Combining this with equation 8 gives

$$
\begin{aligned}
|\log f(1) - \log f(0)| &\leq C_d \max_{\theta \in \Theta} \left\| \sqrt{\bar{F}(\theta)} - \sqrt{\bar{F}'(\theta)} \right\| \int_0^1 \frac{1}{t^d f(1) + (1-t)^d f(0)} dt \\
&\leq C_d \max_{\theta \in \Theta} \left\| \sqrt{\bar{F}(\theta)} - \sqrt{\bar{F}'(\theta)} \right\| \left( \int_0^{1/2} \frac{1}{t^d f(1)} dt + \int_{1/2}^1 \frac{1}{(1-t)^d f(0)} dt \right) \\
&\leq C_d \max_{\theta \in \Theta} \left\| \sqrt{\bar{F}(\theta)} - \sqrt{\bar{F}'(\theta)} \right\| \left( \frac{1}{f(0)} + \frac{1}{f(1)} \right).
\end{aligned}
$$

Combining this with

$$|\tilde{d}_{n,\gamma}(F) - \tilde{d}_{n,\gamma}(F')| \leq \frac{2}{\kappa_{n,\gamma}} |\log f(1) - \log f(0)|$$

almost completes the proof. The final thing to note is that

$$f(0) = \frac{1}{V_\Theta} \int_\Theta \det \left( \mathrm{id}_d/\sqrt{\kappa_{n,\gamma}} + \sqrt{\bar{F}'(\theta)} \right) d\theta \geq \frac{1}{V_\Theta} \int_\Theta \det \left( \sqrt{\bar{F}'(\theta)} \right) d\theta,$$

and similarly for $f(1)$. □

## C  GENERALIZATION BOUND FOR LOG-LIPSCHITZ LOSS FUNCTIONS

In this appendix we prove a generalization of Theorem 4.1 where the loss function is assumed to be log-Lipschitz continuous instead of Lipschitz continuous.

**Theorem C.1.** *Consider the same setting as in Theorem 4.1 with $\epsilon \in (1/\sqrt{n}, 1]$, but the loss function is log-Lipschitz continuous with constant $M_2$ in the first argument with respect to the total variation distance. Then*

$$
\mathbb{P}\left(\sup_{\theta \in \mathcal{B}_\epsilon(\theta^\star)} |R(\theta) - R_n(\theta)| \geq \frac{2M\epsilon}{\sqrt{\kappa_{n,\gamma}}} \log\left(\mathrm{e} + \frac{\sqrt{\kappa_{n,\gamma}}}{M_2\epsilon}\right)\right)
$$
$$
\leq c_d(1+\epsilon\Lambda)^d \cdot \kappa_{n,\gamma}^{\frac{d_{n,\gamma,\epsilon}}{2}} \exp\left(-\frac{2nM^2\epsilon^2}{\kappa_{n,\gamma}B^2}\left(\log\left(\mathrm{e} + \frac{\sqrt{\kappa_{n,\gamma}}}{M_2\epsilon}\right)\right)^2\right),
$$

(9)

*where $M = M_1 M_2$ and $\kappa_{n,\gamma}$ is defined in equation 2.*

To prove Theorem C.1 we need a preparatory lemma.

**Lemma C.2.** *Under the assumption of Theorem C.1, we have for any $\xi \in (0,1)$*

$$
\mathbb{P}\left(\sup_{\theta \in \mathcal{B}_\epsilon(\theta^\star)} |R(\theta) - R_n(\theta)| \geq \xi\right) \leq 2\mathcal{N}^{\mathcal{B}_\epsilon(\theta^\star)}(r) \exp\left(-\frac{n\xi^2}{2B^2}\right),
$$

*where $r = r(\xi)$ is defined as the unique value such that $2M_1 M_2 r \log(\mathrm{e} + \frac{1}{M_2 r}) = \xi/2$.[9]*

*Proof.* Let $r \in \mathrm{P}(\mathcal{X})$ and $q \in \mathrm{P}(\mathcal{Y})$ denote the observed input and output distributions, respectively. Then using the log-Lipschitz assumption of the loss function, we find

$$
|R(\theta_1) - R(\theta_2)|
$$
$$
= \left|\mathbb{E}_{r,q}\left[\ell\big(p(y|x;\theta_1)r(x), q(y)\big)\right] - \mathbb{E}_{r,q}\left[\ell\big(p(y|x;\theta_2)r(x), q(y)\big)\right]\right|
$$
$$
\leq \mathbb{E}_{r,q}\left[\left|\ell\big(p(y|x;\theta_1)r(x), q(y)\big) - \ell\big(p(y|x;\theta_2)r(x), q(y)\big)\right|\right]
$$
$$
\leq M_2 \mathbb{E}_r\left[\|p(y|x;\theta_1)r(x) - p(y|x;\theta_2)r(x)\|_1 \log\left(\mathrm{e} + \frac{1}{\|p(y|x;\theta_1)r(x) - p(y|x;\theta_2)r(x)\|_1}\right)\right]
$$
$$
\leq M_2 \|p(y|x;\theta_1) - p(y|x;\theta_2)\|_\infty \log\left(\mathrm{e} + \frac{1}{\|p(y|x;\theta_1) - p(y|x;\theta_2)\|_\infty}\right)
$$
$$
\leq M_2 M_1 \|\theta_1 - \theta_2\|_\infty \log\left(\mathrm{e} + \frac{1}{\|\theta_1 - \theta_2\|_\infty}\right),
$$

(10)

where the penultimate step uses that $\mathbb{R}_+ \ni x \mapsto x\log(\mathrm{e} + 1/x)$ is monotone together with Hölder's inequality. The final step follows from the Lipschitz continuity assumption of the model. Equivalently we see that

$$
|R_n(\theta_1) - R_n(\theta_2)| \leq M_2 M_1 \|\theta_1 - \theta_2\|_\infty \log\left(\mathrm{e} + \frac{1}{\|\theta_1 - \theta_2\|_\infty}\right).
$$

(11)

Combining equation 10 with equation 11 gives for $S(\theta) := R(\theta) - R_n(\theta)$

$$
|S(\theta_1) - S(\theta_2)| \leq 2M_1 M_2 \|\theta_1 - \theta_2\|_\infty \log\left(\mathrm{e} + \frac{1}{\|\theta_1 - \theta_2\|_\infty}\right).
$$

(12)

Assume that $\mathcal{B}_\epsilon(\theta^\star)$ can be covered by $k$ subsets $B_1, \ldots, B_k$, i.e. $\mathcal{B}_\epsilon(\theta^\star) = B_1 \cup \ldots \cup B_k$. Then, for any $\xi > 0$,

$$
\mathbb{P}\left(\sup_{\theta \in \mathcal{B}_\epsilon(\theta^\star)} |S(\theta)| \geq \xi\right) = \mathbb{P}\left(\bigcup_{i=1}^k \sup_{\theta \in B_i} |S(\theta)| \geq \xi\right) \leq \sum_{i=1}^k \mathbb{P}\left(\sup_{\theta \in B_i} |S(\theta)| \geq \xi\right),
$$

(13)

---

[9] With the convention that $r = \infty$ if $M_1 = 0$ or $M_2 = 0$.

where the inequality is due to the union bound.

Finally, let $k = \mathcal{N}(r)$ and let $B_1, \ldots, B_k$ be balls of radius $r$ centered at $\theta_1, \ldots, \theta_k$ covering $\mathcal{B}_\epsilon(\theta^\star)$. Recalling that by assumption $r = r(\xi)$ is such that $2M_1 M_2 r \log(e + \frac{1}{M_2 r}) = \xi/2$ we find for all $i = 1, \ldots, k$

$$\mathbb{P}\left(\sup_{\theta \in B_i} |S(\theta)| \geq \xi\right) \leq \mathbb{P}\left(|S(\theta_i)| \geq \frac{\xi}{2}\right). \tag{14}$$

To see this recall that since $|\theta - \theta_i| \leq r$, by definition of $r$ and using the monotonicity of $x \mapsto x \log(e + 1/x)$, Inequality equation 12 implies $|S(\theta) - S(\theta_i)| \leq \xi/2$. Hence, if $|S(\theta)| \geq \xi$, it must be that $|S(\theta_i)| \geq \frac{\xi}{2}$. This in turn implies equation 14.

To conclude, we apply Hoeffding's inequality, which yields

$$\mathbb{P}\left(|S(\theta_i)| \geq \frac{\xi}{2}\right) = \mathbb{P}\left(|R(\theta_i) - R_n(\theta_i)| \geq \frac{\xi}{2}\right) \leq 2\exp\left(\frac{-n\xi^2}{2B^2}\right). \tag{15}$$

Combined with equation 13, we obtain

$$\mathbb{P}\left(\sup_{\theta \in \mathcal{B}_\epsilon(\theta^\star)} |S(\theta)| \geq \xi\right) \leq \sum_{i=1}^k \mathbb{P}\left(\sup_{\theta \in B_i} |S(\theta)| \geq \xi\right) \leq \sum_{i=1}^k \mathbb{P}\left(|S(\theta_i)| \geq \frac{\xi}{2}\right)$$
$$\leq 2\mathcal{N}(r) \exp\left(\frac{-n\xi^2}{2B^2}\right),$$

where the second step uses equation 14. The final step follows from equation 15 and by recalling that $k = \mathcal{N}(r)$. $\qquad\square$

*Proof of Theorem C.1.* Choosing $r = \epsilon/\sqrt{\kappa_{n,\gamma}}$ implies

$$\xi = \frac{2M_1 M_2 \epsilon}{\sqrt{\kappa_{n,\gamma}}} \log\left(e + \frac{\sqrt{\kappa_{n,\gamma}}}{M_2 \epsilon}\right) \tag{16}$$

via the relation between $r$ and $\xi$ given in Lemma C.2. Hence Lemma C.2 implies

$$\mathbb{P}\left(\sup_{\theta \in \mathcal{B}_\epsilon(\theta^\star)} |R(\theta) - R_n(\theta)| \geq \xi\right) \leq 2\mathcal{N}^{\mathcal{B}_\epsilon(\theta^\star)}\left(\frac{\epsilon}{\sqrt{\kappa_{n,\gamma}}}\right) \exp\left(-\frac{n\xi^2}{2B^2}\right)$$
$$\leq 2c_d(1 + \epsilon\Lambda)^d \cdot \kappa_{n,\gamma}^{\frac{d_{n,\gamma,\epsilon}}{2}} \exp\left(-\frac{n\xi^2}{2B^2}\right)$$
$$= 2c_d(1 + \epsilon\Lambda)^d \cdot \kappa_{n,\gamma}^{\frac{d_{n,\gamma,\epsilon}}{2}} \exp\left(-\frac{2nM^2\epsilon^2}{\kappa_{n,\gamma}B^2}\left(\log\left(e + \frac{\sqrt{\kappa_{n,\gamma}}}{M_2\epsilon}\right)\right)^2\right),$$

where the second step uses Lemma 4.3 and the final step follows from equation 16. $\qquad\square$

# D  REMARKS ON THE GENERALIZATION ERROR BOUND

Lemma 4.2 implies that $\lim_{n \to \infty} \mathbb{P}(\sup_{\theta \in \mathcal{B}_\epsilon(\theta^\star)} |R(\theta) - R_n(\theta)| \geq \xi) = 0$ for $\xi \in (0, 1)$. As a result, to ensure that the generalization bound in equation 4.1 is meaningful, the right-hand side must vanish as $n \to \infty$. This depends on the problem setting, in particular, on the parameters $\epsilon > 1/\sqrt{n}$, and $\gamma \in (\frac{2\pi \log n}{n}, 1]$. There is some flexibility in choosing these parameters, with a "critical" scaling obtained if $\epsilon = \Omega(1/\log(n))$.[10] In the case where $\epsilon = O(1/n^p)$ for $p < \frac{1}{2}$, the generalization bound gets vacuous for sufficiently large $n$, regardless of the choice of the constant $\gamma \in (\frac{2\pi \log n}{n}, 1]$.[11]

Ideally, the generalization bound from equation 4.1 should be non-vacuous. This occurs if the right-hand side is smaller than one, or equivalently, when the logarithm of the right-hand side is negative.

---

[10] In this case, the two terms in the right-hand side of equation 4.1 balance each other out and the hyperparameter $\gamma$ can control the behaviour of the generalization bound as $n \to \infty$.

[11] Recall that choosing $\gamma$ dependent on $n$ would conflict with the geometric interpretation of the effective dimension (Berezniuk et al., 2020; Abbas et al., 2021).

Table 2 demonstrates that a choice for $\gamma \in (\frac{2\pi \log n}{n}, 1]$ such that the bound remains non-vacuous, is reasonable in practical settings where we set $\gamma = 0.003$. As a result, we plot the accompanying error bound $\xi_n$, the local effective dimension and the logarithm of the right-hand side of equation 4.1 which remains negative, for increasing values of $n$.

Table 2: Evaluation of the generalization bound equation 4.1 for a feedforward neural network trained on MNIST. The model sets $d \approx 10^5$, $\epsilon = 1/\sqrt{n}$, $c_{\Lambda,d} = 2\sqrt{d}$, and $B = M = 1$. Even when setting $\epsilon = 1/\sqrt{n}$, we can still fix $\gamma = 0.003$ such that the generalization bound is non-vacuous, i.e., the RHS of equation 4.1 is $\leq 1$. In fact, the log RHS of equation 4.1 is strongly negative, implying that the RHS is virtually zero. Following equation 4.1, the error bound is given by $\xi_n = 4M\epsilon(\frac{2\pi \log n}{\gamma n})^{1/2} = \frac{4\sqrt{2\pi \log n}}{n\sqrt{\gamma}} \sim 1/n$.

| $n$ | $d_{n,\gamma,\epsilon}$ | $\xi_n$ | log RHS of equation 4.1 |
|---|---|---|---|
| $5 \times 10^5$ | 23474 | 0.00132 | $-98507$ |
| $10^6$ | 25285 | 0.00068 | $-91345$ |
| $2 \times 10^6$ | 27594 | 0.00034 | $-79921$ |
| $5 \times 10^6$ | 31106 | 0.00014 | $-59307$ |
| $10^7$ | 33933 | 0.00007 | $-40316$ |

## E    EXPERIMENTAL SETUP

Here, we explain the models and techniques used for the experiments in this study. All models constituted fully-connected feedforward neural networks with leaky relu activation functions. We used 2 hidden layers for all architectures, but varied the number of neurons per layer depending on the experiment. Training was done with stochastic gradient descent, with batch sizes equal to $50$. In the instances where the CIFAR10 data set was used, we performed a standard transformation of the data by normalizing and cropping from the center. For more details on this transformation, see (Zhang et al., 2021).

### E.1    INCREASING MODEL SIZE

In Figures 1(a) and 2(a) we train feedforward neural networks on the MNIST and CIFAR10 data sets respectively. In both cases, we plot the model size on the x-axis and incrementally increase the number of neurons in both hidden layers, thereby increasing the number of parameters in the model. For MNIST, we do not need to train very large models to achieve zero training error, thus, we vary the number of parameters from $2 \times 10^4$ to $10^5$. On the other hand, for CIFAR10, we train models with parameters ranging from $5 \times 10^5$ to $10^7$. In MNIST, the training and test split is 60000 and 10000 images respectively. For CIFAR10, it is 50000 and 10000.

We train every model for 200 epochs and plot the resulting generalization errors, approximated by the test error. We also plot the normalized local effective dimension for every model using the trained parameter set $\theta^*$. For this, we use $n = 6 \times 10^4$ (which is the size of both training sets) and set $\gamma = 1$.

In both Figures, we repeat the entire experiment 10 times with different parameter initialization and plot the average generalization error and average normalized local effective dimension over these 10 trials, with error bars depicting $\pm 1$ standard deviation above and below the mean values. As expected, the local effective dimension declines along with the generalization error in both shallow (MNIST) and deep (CIFAR10) regimes.

### E.2    RANDOMIZATION EXPERIMENT

In Figures 1(b) and 2(b) we train models on the MNIST and CIFAR10 data sets respectively. In this experiment, both models are fixed to $d \approx 10^5$ for MNIST and $d \approx 10^7$ for CIFAR10. What we vary is the proportion of training labels that are replaced with random labels (as originally done in (Zhang et al., 2021)). We begin by randomizing $20\%$ of the training labels as shown on the x-axis. We train the models to zero training error or terminate after 600 epochs and plot the resulting generalization

error and effective dimension. Thereafter, we increase the proportion of random labels in increments of 20%, until 100% randomization and plot the generalization error and normalized local effective dimension after training, each time.

This entire process is repeated 10 times with different parameter initialization and we plot the average generalization error and average normalized local effective dimension over increasing label randomization. Unsurprisingly, the generalization error increases as we increase the level of randomization since the network is essentially learning to fit more and more noise and thus, does not generalize well. Interestingly, the local effective dimension moves in line with this trend and increases over increasing randomization too. This could be interpreted as the network requiring more and more parameters to forcefully fit the increasing noise levels that would not naturally occur. Thus, the local effective dimension captures the correct generalization behaviour, even in this artificial set up where most capacity measures fail to explain generalization performance.

While in the case of noisy labels, deep neural networks seem to overfit regardless of how deep you make them, it is worthwhile to mention other progress in understanding deep learning through experiments aiming to probe the nature of overfitting. One particular analysis involves the study of memorization of overparameterized models which produces a "double descent" curve rather than the traditional U-shaped risk curve (Belkin et al., 2019). The details of the double descent phenomenon are very subtle and depend on several interrelating factors such as the distribution of data, the optimizer used (i.e. the trained parameters) and the notion of capacity employed. Given that the local effective dimension accounts for these factors through the nature of its definition and it is able to capture overfitting in an overparameterized regime induced through artificial noise in the labels, we postulate that the local effective dimension through training would also track the double descent risk curve accordingly. Thus far, we have conducted various experiments that calculate the local effective dimension after training is complete, but the measure should intuitively inform us about local information around any set of parameters. Thus, one could also extend the analysis to look at the local effective dimension throughout training, which should mirror the double descent phenomenon containing various regimes of under- and overfitting as in the work of (Maddox et al., 2020) who employ a different notion of effective dimension.

### E.3    ESTIMATING THE LOCAL EFFECTIVE DIMENSION

In all calculations involving the local effective dimension, there are two assumptions made. First, we use a fixed parameter set $\theta^*$ chosen after training to estimate the local effective dimension and assume it is a good approximation of the average of sampling in an $\epsilon$-ball around $\theta^*$. In other words, we ignore the integral over $\mathcal{B}_\epsilon(\theta^\star)$ in Definition 3.4 and simply use the trained parameter set to compute the local effective dimension. In Table 3 we check the sensitivity of the local effective dimension by comparing this "midpoint" approximation to sampling in an $\epsilon$-ball around $\theta^*$ and conclude that the approximation is sufficiently close and thus helps reduce computational time. We use the more efficient reformulation from Remark 3.3 to calculate the local effective dimension with multiple samples and take $\epsilon = 1/\sqrt{n}$.

Table 3: Evaluation of the local effective dimension for a feedforward neural network trained on CIFAR10 with $d \approx 10^7$. We plot values for the normalized local effective dimension $\bar{d}$ calculated with increasing samples from an $\epsilon$-ball around the trained $\theta^*$, where $\epsilon = 1/\sqrt{n}$. The midpoint approximation uses the single $\theta^*$ after training. For completeness, we include the unnormalized local effective dimension $d_{n,\gamma,\epsilon}$ and the average of the $z(\theta)$ values generated from each sample as defined in equation 4.

| Samples | Midpoint | 50 | 100 | 200 | 500 | 1000 |
|---|---|---|---|---|---|---|
| $\bar{d}$ | 0.21815588 | 0.21815937 | 0.21816021 | 0.21815975 | 0.218159262 | 0.21815838 |
| $d_{n,\gamma,\epsilon}$ | 2189499.62 | 2189534.68 | 2189543.14 | 2189538.53 | 2189533.61 | 2189524.72 |
| Average of $z(\theta)$ | N/A | 7407283.05 | 7407306.34 | 7407286.62 | 7407213.06 | 7407150.42 |

The second assumption made is that the Fisher information matrix can be approximated by the empirical Fisher information matrix. From (Liao et al., 2018), we acknowledge that this assumption does not always necessarily hold. Thus, the continuity statement from Proposition 3.2 becomes

relevant to ensure errors introduced by the estimate of the empirical Fisher do not strongly propagate in the calculation of the local effective dimension.

Through the empirical Fisher information assumption, we further exploit work done in (Martens & Grosse, 2015) and use the Kronecker-Factored Approximation (K-FAC) Fisher matrix in the estimation of the effective dimension. The K-FAC Fisher allows us to estimate the eigenvalues of the empirical Fisher information much more efficiently for large models. We slightly extend the PyTorch (Paszke et al., 2019) implementation developed in (George, 2021). We refer the interested reader to (Martens & Grosse, 2015) for more details. The K-FAC estimate constitutes several block matrices which comprise a diagonal block estimate of the empirical Fisher estimate. The block matrices relate to the hidden layers used in a neural network model. Conveniently, these block matrices can be further factorized into a tensor product of two smaller matrices. Thus, to calculate the eigenvalues of the K-FAC Fisher, it suffices to compute the eigenvalues of the block matrices, and thereby take advantage of their tensor decomposition. We extend the PyTorch K-FAC implementation from (George, 2021) to include a function that computes all the eigenvalues. Thereafter, the estimation of the local effective dimension follows from equation 4. Due to these approximations and the code implementation in (George, 2021), we would like to highlight that computational bottleneck lies solely with training the model when $d$ is very large. The memory and time overhead for computing the local effective dimension with the K-FAC Fisher is highly efficient and could, for example, run on a laptop with modest memory for a model with $d = 10000000$ in just few minutes.

### E.4 ADDITIONAL EXPERIMENTS

In order to validate the robustness of our results with different optimizers, we have conducted the same experiment as in Figure 2 using the ADAM optimizer. As was the case with stochastic gradient descent, we see that the relationship between generalization and the local (normalized) effective dimension still holds.

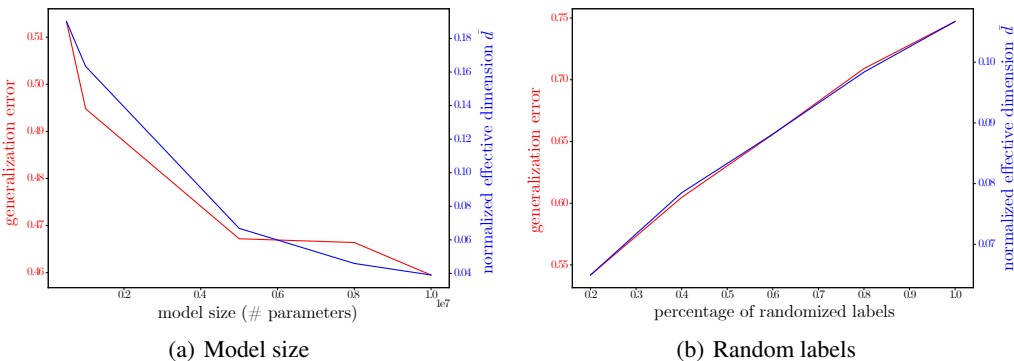

| (a) Model size | (b) Random labels |
|---|---|

Figure 3: **CIFAR10 with ADAM** (a) Normalized local effective dimension and generalization error plotted over different model sizes. The number of parameters required to train CIFAR10 is far greater than the number of training samples ($n = 50000$). We observe that the local effective dimension moves in line with the declining generalization error as the model is made larger, even with training using a different optimizer (ADAM as opposed to SGD in Figure 3. (b) Normalized local effective dimension and generalization error over different percentages of randomized labels on the training data. Here, the model size is fixed to $d \approx 10^7$ and training using the ADAM optimizer.

