# OpenReview forum: "Effective dimension of machine learning models"
_ICLR.cc/2023/Conference — Submitted to ICLR 2023_

### Official Review · Reviewer_Ec2u · 2022-10-21

**Confidence:** 4
**Correctness:** 2
**Technical Novelty And Significance:** 2
**Empirical Novelty And Significance:** 2
**Recommendation:** 3

**Clarity, Quality, Novelty And Reproducibility:**

The notion of the local effective dimension is derived from the previous notion of the global effective dimension, and the authors try to generalize the global notion into a local version.

**Strength And Weaknesses:**

Strength:
1. The authors propose a new measure called the local effective dimension. The new notion has some interesting properties, e.g., algorithm-dependent.
2. The authors validate the new notion empirically and find that it works well in practice.
3. This paper considers a very important topic generalization.

Weakness:
1. [Major concern]  Maybe I lose some details. I am not sure that the generalization bound holds. In this paper, the local version means that the bounds are derived around $\theta^*$. Although the authors claim that $\theta^*$ is a *fixed* parameter, in practice, it is usually the trained parameter. Therefore, $\theta^*$ is data- and algorithm-dependent and therefore is related to the training set.
So, when generalizing the notion from the global version to the local version, some arguments may dramatically fail due to the dependency of $\theta^*$. For example, in Lemma 4.2, the authors use covering number-type techniques, but it may not work around the trained parameter $\theta^*$ (because the reference point is all around $\theta^*$ and therefore, one cannot apply Hoeffding). Could the author explain more about that?

2. In the experiment, the authors show that the local effective dimension is correlated to generalization error. However, the theoretical results seem to imply that the local effective dimension is on a power exponent. I am not sure that this experiment can validate the theorem.

3. Since there is a sharpness term in estimating effective dimension. It seems that the sharpness term cannot be directly derived in practice. Did the authors consider the estimation error on the sharpness when deriving the bound?


**Summary Of The Paper:**

This paper aims to analyze generalization via a notion called local effective dimension, which captures the local properties of a function class. The notion of the local effective dimension is derived from the previous notion of the global effective dimension, and the authors try to generalize the global notion into a local version. As the authors claimed, the new local version has some interesting benefits, such as being algorithm-dependent. The authors finally validate the local effective dimension both theoretically and empirically.

Although the whole idea is interesting, I feel puzzled by the theoretical part (see weaknesses). Maybe I missed some details, and the authors can point them out. I do not mind increasing my ratings if these concerns are answered properly.



**Summary Of The Review:**

It would be nice to see the author response on the correctness of the paper.

---

### Official Review · Reviewer_g1fU · 2022-10-24

**Confidence:** 4
**Correctness:** 4
**Technical Novelty And Significance:** 3
**Empirical Novelty And Significance:** 2
**Recommendation:** 5

**Clarity, Quality, Novelty And Reproducibility:**

The paper is well-written and the results are clear.

The main issue with the results are 1) the intuition behind the results hasn't been discussed. 2) the connection to other capacity measures is not clear.

The authors have some preliminary experiments which show that the generalization error correlates with the effective dimension.

**Strength And Weaknesses:**

Questions:

1- In the definition of the Fisher Information, is the expectation over the data generating distribution? it is not very clear from the paper.

2- The definition of the effective dimension, there are lots of integrals. What are the necessary and sufficient conditions for the existence of the integrals?

3- In the case of deterministic classifiers, for instance VC classes for binary classification, how can we define the effective dimension?

4- Why is the Lebesgue msr chosen as the base measure for measuring the volume?

5- The authors should provide more intuition behind their definition.

6- what is the connection of the effective dimension to other well-known measures such as Radamacher complexity and VC dimension?

7- The authors miss an important result on the uniform convergence:

 [1] Uniform convergence may be unable to explain generalization in deep learning Vaishnavh Nagarajan, J. Zico Kolter

It is interesting to discuss the counter examples in [1] in your paper.

8- What are some examples for which we can theoretically find the effective dimension?


**Summary Of The Paper:**

This paper introduces a new capacity measure for characterizing the generalization called "global effective dimension".

The definition is based on Fisher information. The authors also show that this measure can provide the error of uniform convergence.

**Summary Of The Review:**

This paper presents a new notion for characterizing the error of uniform convergence. The numerical results show that unlike VC dimension and Radamacher complexity this measure can correlate well with the generalization error. The main issue with the paper is that there is not intuitive explanation of this measure. Also, its connections with other notions such as VC dimension and Radamacher complexity is not clear.

---

### Official Review · Reviewer_c3zn · 2022-10-25

**Confidence:** 3
**Correctness:** 2
**Technical Novelty And Significance:** 2
**Empirical Novelty And Significance:** 2
**Recommendation:** 3

**Clarity, Quality, Novelty And Reproducibility:**

The paper is clearly written for the most part.

The methodology is novel to the best of my knowledge, although I am not a theorist and may not have the most thorough knowledge of the latest results of learning theory.

The results appear to be reproducible.

Unfortunately, the quality is not high enough for ICLR, given the lack of benchmarking, the lack of characterization of tightness of bounds and the numerous misstatements regarding VC dimension.

I suspect this line of investigation may well be worth pursuing, but the research is not ready for publication and the authors should spend some time double-checking their understanding of VC theory.

**Strength And Weaknesses:**

The writing is reasonably clear, for the most part. This is certainly a strength of the paper.

My expertise is not primarily in learning theory although I did publish a little bit on learning theory early in my career, so I know VC dimension better than some of the more recent competing generalization measures the paper discusses. It concerns me greatly that there are several statements regarding VC theory (in Table 1 with the x's and checks and in the text) that are just flatly wrong.

1) Table 1 claims VC dimension is not "scale invariant" and describes scale invariance as being insensitive to inconsequential transformations such as multiplying a neural net's weights by a constant. Bounds on the VC dimension of a neural net with a given architecture can be obtained which hold for the entire class of functions implementable by any choice of weights for the architecture, so I fail to see how VC dimension in any sense fails to be "scale invariant".

2) Table 1 claims VC dimension is not data dependent, but there is a related measure, VC entropy, which depends on the expected number of dichotomies for a given input distribution (see e.g. https://www.shivani-agarwal.net/Teaching/E0370/Aug-2011/Lectures/3.pdf)  so the VC framework is in fact capable of taking into account data distributions.

3) Page 2 and Table 1 both claim VC dimension assumes access to infinite data. I don't understand this claim at all , but it is simply wrong. VC theory provides a generalization bound for a given function class and a given finite training sample size N. It may indeed tend to be a loose bound when N is not very large, but there is no meaningful sense in which it assumes access to infinite data.

4) Table 1 claims VC dimension does not provide "efficient evaluation", yet there are well known bounds on the VC dimension of neural nets as a function of hidden units and connectivity which are trivial to compute, so this claim is not correct, either.

To be honest, I did not have time to go through every line of the derivation of local effective dimension in this paper very carefully, but given the volume of incorrect statements about VC dimension here, I do not have high confidence in the correctness of the core results of the paper.

My other concerns are as follows:

1) The experimental results merely show that local effective dimension correlates with test error. They do appear to do any benchmarking, i.e., they do not show that it correlates with test error better than competing methods .

2) The discussion section says it would be beneficial to investigate the tightness of the bounds, conceding that no such investigations of tightness are presented here. It's hard to judge the significance of the results without any such measure of tightness.



**Summary Of The Paper:**

A complexity measure called the local effective dimension is proposed for bounding the generalization error of a supervised learning model. Local effective dimension is related to the Fisher information matrix constrained to a particular region of parameter space, with that parameter region intended to capture the fact that only a certain area of parameter space is explored during training. A list of desirable properties of generalization bounds is presented and it is claimed that local effective dimension satisfies all the desirable properties, unlike previously available generalization theory methodologies.  Theoretical results are presented showing that local effective dimension provides a bound on generalization error. Experimental results using MNIST and CIFAR10 show that local effective dimension empirically correlates with generalization error.

**Summary Of The Review:**

A generalization measure is presented which is related to Fisher information and which appears to correlate with test error empirically, given some neural net experiments with image data. However, numerous misstatements are made about VC theory, the tightness of the bounds presented is unclear and the empirical results are not well benchmarked.

---

### Official Review · Reviewer_x2eo · 2022-11-03

**Confidence:** 3
**Correctness:** 3
**Technical Novelty And Significance:** 4
**Empirical Novelty And Significance:** 3
**Recommendation:** 5

**Clarity, Quality, Novelty And Reproducibility:**

The paper appears excellent in each of these regards: Clarity, novelty, and reproducibility.

**Strength And Weaknesses:**

\+ The review of prior work is highly readable and interesting. It does a good job of categorizing and describing the intuitions behind prior work, rather than merely providing a large list of papers.

\+ The paper is well-written and clear.

– Section 2 is confusing, in parts. The discussion at the beginning would be clearer if the authors provided a better description of what is meant by an "estimate of error."  Also, the discussion of Fisher information in Section 2.2 is completely unmotivated.

– The concept of effective dimension would appear to require knowledge of the distribution of the data. For example, consider a simple logistic regression model with k+1 parameters, where k is the number of predictor variables. If each variable X_2...X_k are copies of X_1, then the potential for the model to fit an arbitrary target variable Y is much smaller than if X_2...X_k are independent.

– The results in Figure 2 are less impressive than they might appear. What the graphs show clearly is that LED decreases with the number of parameters and increases with the percentage of randomized labels (as does generalization error). By adjusting scales, LED has been made to match the beginning and end points of the generalization error curve. To their credit, the authors don't over-interpret this evidence ("...the local effective dimension seems to move in line with the generalization error." and "Whilst the search for a good capacity measure continues, we believe that the local effective dimension serves as a promising candidate."). However, some readers may not be so thoughtful.

– The authors provide only preliminary empirical evidence of the utility of LED. The experiments that are provided are interesting. However, the results are given for only two data sets and only one type of model and training method. Sample size is not varied, nor are characteristics of the data (e.g., number of variables or level of randomization of the predictor variables).

**Summary Of The Paper:**

The authors propose "local effective dimension" (LED) as a measure of the capacity of a machine learning model. They show that it appears to correlate with generalization error on standard data sets and they prove that it bounds the generalization error. They compare their proposed measure to a number of others, including VC-dimension and Rademacher complexity.

**Summary Of The Review:**

The paper is very promising in terms of nearly all aspects. It addresses an important topic, it proposes a reasonable candidate measure, it contains substantial technical justification for the measure, and it empirically evaluates the performance of the measure. However, the empirical evaluation is sparse, and readers have insufficient evidence to conclude whether the measure is worth pursuing. The authors are clearly in the best position to provide that evidence, but the current paper does not provide it. The paper would be acceptable given a far more extensive empirical evaluation (and assuming, of course, that the results of that evaluation continue to show the measure to be effective).

---

### Decision · Program_Chairs · 2023-01-20

**Decision:**

Reject

**Justification For Why Not Higher Score:**

N/A

**Justification For Why Not Lower Score:**

N/A

**Metareview: Summary, Strengths And Weaknesses:**

This paper proposes a "local effective dimension" (LED) as a measure of the capacity of a machine learning model and shows that it appears to correlate with generalization error on standard data sets and proves that it bounds the generalization error. They compare their proposed measure to VC-dimension and Rademacher complexity.

The meta-reviewer has carefully read the paper and all reviews and agrees with the reviewers' concerns. Although the writing of this paper is clear, the definition of the effective dimension is unclear with certain assumptions such as the data distribution that is difficult to obtain. The evaluation of the LED was on MNIST and CIFAR10, which seem to be limited to justify the paper claim. The authors didn't respond to the reviews. The meta-reviewer recommends a rejection.